# Spatial Graph Attention and Curiosity-driven Policy for Antiviral Drug Discovery

**Yulun Wu** [† §]
yulun_wu@berkeley.edu

**Mikaela Cashman** [¶ §]
cashmanmm@ornl.gov

**Nicholas Choma** [‡ §]
njchoma@lbl.gov

**Érica T. Prates** [¶ §]
teixeiraprae@ornl.gov

**Verónica Melesse Vergara** [‖ ¶]
vergaravg@ornl.gov

**Manesh Shah** [‖]
shahmb@ornl.gov

**Andrew Chen** [‡ §]
adchen@lbl.gov

**Austin Clyde** [††]
aclyde@uchicago.edu

**Thomas S. Brettin** [§§]
brettin@anl.gov

**Wibe A. de Jong** [‡ §]
wadejong@lbl.gov

**Neeraj Kumar** [‡‡ §]
neeraj.kumar@pnnl.gov

**Martha S. Head** [¶ §]
headms@ornl.gov

**Rick L. Stevens** [†† §§]
stevens@anl.gov

**Peter Nugent** [‡ §]
penugent@lbl.gov

**Daniel A. Jacobson** [‖ ¶ §]
jacobsonda@ornl.gov

**James B. Brown** [† ‡ §]
jbbrown@lbl.gov

## Abstract

We developed Distilled Graph Attention Policy Network (DGAPN), a reinforcement learning model to generate novel graph-structured chemical representations that optimize user-defined objectives by efficiently navigating a physically constrained domain. The framework is examined on the task of generating molecules that are designed to bind, noncovalently, to functional sites of SARS-CoV-2 proteins. We present a spatial Graph Attention (sGAT) mechanism that leverages self-attention over both node and edge attributes as well as encoding the spatial structure — this capability is of considerable interest in synthetic biology and drug discovery. An attentional policy network is introduced to learn the decision rules for a dynamic, fragment-based chemical environment, and state-of-the-art policy gradient techniques are employed to train the network with stability. Exploration is driven by the stochasticity of the action space design and the innovation reward bonuses learned and proposed by random network distillation. In experiments, our framework achieved outstanding results compared to state-of-the-art algorithms, while reducing the complexity of paths to chemical synthesis.

## 1 Introduction

This work aims to address the challenge of establishing an automated process for the design of objects with connected components, such as molecules, that optimize specific properties. Achieving this goal is particularly desirable in drug development and materials science, where manual discovery remains a time-consuming and expensive process (Hughes et al., 2011; Schneider et al., 2020). However, there are two major difficulties that have long impeded rapid progress. Firstly, the chemical space is discrete and massive (Polishchuk et al., 2013), presenting a complicated environment for an Artificial Intelligence (AI) approach to efficiently and effectively explore. Secondly, it is not trivial to compress such connected objects into feature representations that preserve most of the information, while also being highly computable for Deep Learning (DL) methods to exploit.

We introduce Distilled Graph Attention Policy Network (DGAPN), a framework that advances prior work in addressing both of these challenges. We present a Reinforcement Learning (RL) architecture that is efficiently encouraged to take innovative actions with an environment that is able to construct a

---

† University of California, Berkeley, § National Virtual Biotechnology Laboratory, US Department of Energy, ‡ Lawrence Berkeley National Laboratory, ¶ Oak Ridge National Laboratory, ‖ University of Tennessee, Knoxville, †† University of Chicago, §§ Argonne National Laboratory, ‡‡ Pacific Northwest National Laboratory

dynamic and chemically valid fragment-based action space. We also propose a hybrid Graph Neural Network (GNN) that comprehensively encodes graph objects' attributes and spatial structures in addition to adjacency structures. The following paragraphs discuss how we addressed limitations of prior work and its relevance to antiviral drug discovery. For more descriptions of key prior methodologies that we used as benchmarks in this paper, see Section 4.

**Graph Representation Learning** Despite their spatial efficiency, string representation of molecules acquired by the simplified molecular-input line-entry system (SMILES) (Weininger, 1988) suffers from significant information loss and poor robustness (Liu et al., 2017). Graph representations have become predominant and preferable for their ability to efficiently encode an object's scaffold structure and attributes. Graph representations are particularly ideal for RL since intermediate representations can be decoded and evaluated for reward assignments. While GNNs such as Graph Convolutional Networks (GCN) (Kipf & Welling, 2016) and Graph Attention Networks (GAT) (Veličković et al., 2017) have demonstrated impressive performance on many DL tasks, further exploitation into richer information contained in graph-structured data is needed to faithfully represent the complexity of chemical space (Morris et al., 2019; Wang et al., 2019; Chen et al., 2020). In this work, we made improvements to previous studies on attributes encoding and structural encoding. For structural encoding, previous studies have covered adjacency distance encoding (Li et al., 2020), spatial cutoff (Pei et al., 2020) and coordinates encoding (Schütt et al., 2017; Danel et al., 2020). Our work presents an alternative approach to spatial structure encoding similar to Gilmer et al. (2017) which do not rely on node coordinates, but different in embedding and updating scheme. Distinct from Danel et al. (2020) and Chen & Chen (2021), we extended attentional embedding to be edge-featured, while still node-centric for message passing efficiency.

**Reinforcement Learning** A variety of graph generative models have been used in prior work, predominantly Variational Autoencoders (VAE) (Simonovsky & Komodakis, 2018; Samanta et al., 2020; Liu et al., 2018; Ma et al., 2018; Jin et al., 2018) and Generative Adversarial Networks (GAN) (De Cao & Kipf, 2018). While some of these have a recurrent structure (Li et al., 2018; You et al., 2018b), RL and other search algorithms that interact dynamically with the environment excel in sequential generation due to their ability to resist overfitting on training data. Both policy learning (You et al., 2018a) and value function learning (Zhou et al., 2019) have been adopted for molecule generation: however, they generate molecules node-by-node and edge-by-edge. In comparison, an action space consisting of molecular fragments, i.e., a collection of chemically valid components and realizable synthesis paths, is favorable since different atom types and bonds are defined by the local molecular environment. Furthermore, the chemical space to explore can be largely reduced. Fragment-by-fragment sequential generation has been used in VAE (Jin et al., 2018) and search algorithms (Jin et al., 2020; Xie et al., 2021), but has not been utilized in a deep graph RL framework. In this work, we designed our environment with the Chemically Reasonable Mutations (CReM) (Polishchuk, 2020) library to realize a valid fragment-based action space. In addition, we enhanced exploration by employing a simple and efficient technique, adapting Random Network Distillation (RND) (Burda et al., 2018) to GNNs and proposing surrogate innovation rewards for intermediate states during the generating process.

**Antiviral Drug Discovery — A Timely Challenge** The severity of the COVID-19 pandemic highlighted the major role of computational workflows to characterize the viral machinery and identify druggable targets for the rapid development of novel antivirals. Particularly, the synergistic use of DL methods and structural knowledge via molecular docking is at the cutting edge of molecular biology — consolidating such integrative protocols to accelerate drug discovery is of paramount importance (Yang et al., 2021; Jeon & Kim, 2020; Thomas et al., 2021). Here we experimentally examined our architecture on the task of discovering novel inhibitors targeting the SARS-CoV-2 non-structural protein endoribonuclease (NSP15), which is critical for viral evasion of host defense systems (Pillon et al., 2021). Structural information about the putative protein-ligand complexes was integrated into this framework with AutoDock-GPU (Santos-Martins et al., 2021), which leverages the GPU resources from leadership-class computing facilities, including the Summit supercomputer, for high-throughput molecular docking (LeGrand et al., 2020). We show that our results outperformed state-of-the-art generation models in finding molecules with high affinity to the target and reasonable synthetic accessibility.

## 2    PROPOSED METHOD

### 2.1    ENVIRONMENT SETTINGS

In the case of molecular generation, single-atom or single-bond additions are often not realizable by known biochemical reactions. Rather than employing abstract architectures such as GANs to suggest synthetic accessibility, we use the chemical library CReM (Polishchuk, 2020) to construct our environment such that all next possible molecules can be obtained by one step of interchanging chemical fragments with the current molecule. This explicit approach is considerably more reliable and interpretable compared to DL approaches. A detailed description of the CReM library can be found in Appendix B.1.

The generating process is formulated as a Markov decision problem (details are given in Appendix A). At each time step $t$, we use CReM to sample a set of valid molecules $\boldsymbol{v}_{t+1}$ as the candidates for the next state $s_{t+1}$ based on current state $s_t$. Under this setting, the transition dynamics are deterministic, set $A$ of the action space can be defined as equal to $S$ of the state space, and action $a_t$ is induced by the direct selection of $s_{t+1}$. With an abuse of notation, we let $r(s_{t+1}) := r(s_t, a_t)$.

### 2.2    SPATIAL GRAPH ATTENTION

We introduce a graph embedding mechanism called Spatial Graph Attention (sGAT) in an attempt to faithfully extract feature vectors $\boldsymbol{h}_t \in \mathbb{R}^{d_h}$ representing graph-structured objects such as molecules. Two different types of information graphs constructed from a connected object are heterogeneous and thus handled differently in forward passes as described in the following sections. See Figure 1 for an overview.

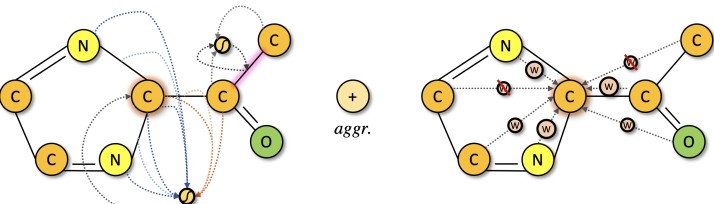

Figure 1: Overview of Spatial Graph Attention defined by equations (1) to (4). Highlighted nodes and edge are the examples undergoing forward propagation. The attention mechanism is node centric: nodes are embedded leveraging information from adjacent nodes and edges (different colors of dash lines denote different attentions); edges are embedded leveraging information from adjacent nodes. Spatial information is separately encoded according to sparsified inverse distance matrix (red crosses represent weights that are omitted) and embedded with such attention mechanism. The two hidden representations acquired respectively are aggregated at the end of each layer.

#### 2.2.1    ATTENTION ON ATTRIBUTION GRAPHS

The attribution graph of a molecule with $n$ atoms and $e$ bonds is given by the triple $(\boldsymbol{A}, \boldsymbol{N}, \boldsymbol{E})$, where $\boldsymbol{A} \in \{0, 1\}^{n \times n}$ is the node adjacency matrix, $\boldsymbol{N}$ is the node attribution matrix of dimension $n \times d_n$ and $\boldsymbol{E}$ is the edge attribution matrix of dimension $e \times d_e$. Each entry $a_{ij}$ of $\boldsymbol{A}$ is 1 if a bond exists between atom $i$ and $j$, and 0 otherwise. Each row vector $\boldsymbol{n}_i$ of $\boldsymbol{N}$ is a concatenation of the properties of atom $i$, including its atomic number, mass, etc., with the categorical properties being one-hot encoded. $\boldsymbol{E}$ is formed similar to $\boldsymbol{N}$, but with bond attributes. We denote a row vector of $\boldsymbol{E}$ as $\boldsymbol{e}_{ij}$ if it corresponds to the bond between atom $i$ and $j$.

We proceed to define a multi-head forward propagation that handles these rich graph information: let $h_{\boldsymbol{n}_k} \in \mathbb{R}^{1 \times d_{h_n}}$ denote a given representation for $\boldsymbol{n}_k$, $h_{\boldsymbol{e}_{ij}} \in \mathbb{R}^{1 \times d_{h_e}}$ denote a representation for $\boldsymbol{e}_{ij}$, then the $m$-th head attention $\alpha_{ij}^m$ from node $j$ to node $i$ ($i \neq j$) is given by

$$\alpha_{ij}^m = softmax_j \left( \bigcup_{k:\, a_{ik}=1} \left\{ \sigma([h_{\boldsymbol{n}_i} \boldsymbol{W}_{n,m} \parallel h_{\boldsymbol{e}_{ik}} \boldsymbol{W}_{e,m} \parallel h_{\boldsymbol{n}_k} \boldsymbol{W}_{n,m}] \cdot att_m{}^T) \right\} \right) \qquad (1)$$

where $softmax_j$ is the softmax score of node $j$; $\|$ is column concatenation; $\sigma$ is some non-linear activation; $\boldsymbol{W}_{n,m} \in \mathbb{R}^{d_{h_n} \times d_{w_n}}$, $\boldsymbol{W}_{e,m} \in \mathbb{R}^{d_{h_e} \times d_{w_e}}$ are the $m$-th head weight matrices for nodes and edges respectively; $att_m \in \mathbb{R}^{1 \times (2d_{w_n} + d_{w_e})}$ is the $m$-th head attention weight. The representations after a feed-forward operation are consequently given as follow:

$$h'_{\boldsymbol{n}_i} = aggr_{1 \leq m \leq n_m} \left\{ \sigma \left( \left( \sum_{j:\, a_{ij}=1} \alpha_{ij}^m \cdot h_{\boldsymbol{n}_j} + h_{\boldsymbol{n}_i} \right) \boldsymbol{W}_{n,m} \right) \right\} \tag{2}$$

$$h'_{\boldsymbol{e}_{ij}} = aggr_{1 \leq m \leq n_m} \left\{ \sigma \left( \left[ h_{\boldsymbol{n}_i} \boldsymbol{W}_{n,m} \,\|\, h_{\boldsymbol{e}_{ij}} \boldsymbol{W}_{e,m} \,\|\, h_{\boldsymbol{n}_j} \boldsymbol{W}_{n,m} \right] \cdot \boldsymbol{W}_{h,m} \right) \right\} \tag{3}$$

where $\boldsymbol{W}_{h,m} \in \mathbb{R}^{(2d_{w_n} + d_{w_e}) \times d_{w_e}}$; $n_m$ is the total number of attention heads and $aggr$ denotes an aggregation method, most commonly $mean$, $sum$, or $concat$ (Hamilton et al., 2017). We note that we have not found significant difference across these methods and have used $mean$ for all aggregations in our experiments. In principle, a single-head operation on nodes is essentially graph convolution with the adjacency matrix $\hat{\boldsymbol{A}} = \widetilde{\boldsymbol{A}} + \boldsymbol{I}$ where $\widetilde{\boldsymbol{A}}$ is attention-regularized according to (1). This approach sufficiently embeds edge attributes while still being a node-centric convolution mechanism, for which efficient frameworks like Pytorch-Geometric (Fey & Lenssen, 2019) have been well established.

### 2.2.2 SPATIAL CONVOLUTION

In addition to attributions and logical adjacency, one might also wish to exploit the spatial structure of an graph object. In the case of molecular docking, spatial structure informs the molecular volume and the spatial distribution of interaction sites — shape and chemical complementarity to the receptor binding site is essential for an effective association.

Let $\boldsymbol{G} = \left( d_{ij}^{-1} \right)_{i,j \leq n}$ be the inverse distance matrix where $d_{ij}$ is the Euclidean distance between node $i$ and $j$ for $\forall i \neq j$, and $d_{ii}^{-1} := 0$. $\boldsymbol{G}$ can then be seen as an adjacency matrix with weighted "edge"s indicating nodes' spatial relations, and the forward propagation is thus given by

$$\boldsymbol{H}''_n = \sigma \left( \left( \widetilde{\boldsymbol{D}}^{-\frac{1}{2}} \widetilde{\boldsymbol{G}} \widetilde{\boldsymbol{D}}^{-\frac{1}{2}} + \boldsymbol{I} \right) \boldsymbol{H}_n \boldsymbol{W}_n \right) \tag{4}$$

where $\widetilde{\boldsymbol{G}}$ is optionally sparsified and attention-regularized from $\boldsymbol{G}$ to be described below; $\widetilde{\boldsymbol{D}} = diag_{1 \leq i \leq n} \left\{ \sum_{j=1}^n \widetilde{G}_{ij} \right\}$; $\boldsymbol{H}_n$ is the row concatenation of $\{h_{\boldsymbol{n}_i}\}_{1 \leq i \leq n}$; $\boldsymbol{W}_n \in \mathbb{R}^{d_{h_n} \times d_{w_n}}$ is the weight matrix. In reality, $\boldsymbol{G}$ induces $O(n)$ of convolution operations on each node and can drastically increase training time when the number of nodes is high. Therefore, one might want to derive $\widetilde{\boldsymbol{G}}$ by enforcing a cut-off around each node's neighborhood (Pei et al., 2020), or preserving an $O(n)$ number of largest entries in $\boldsymbol{G}$ and dropping out the rest. In our case, although the average number of nodes is low enough for the gather and scatter operations (GS) of Pytorch-Geometric to experience no noticeable difference in runtime as node degrees scale up (Fey & Lenssen, 2019), the latter approach of sparsification was still carried out because we have discovered that proper cutoffs improved the validation loss in our supervised learning experiments. If one perceives the relations between chemical properties and spatial information as more abstract, $\boldsymbol{G}$ should be regularized by attention as described in (1), in which case the spatial convolution is principally fully-connected graph attention with the Euclidean distance as a one-dimensional edge attribution.

### 2.3 GRAPH ATTENTION POLICY NETWORK

In this section we introduce Graph Attention Policy Network (GAPN) that is tailored to environments that possess a dynamic range of actions. Note that $\rho(\cdot|s_t, a_t)$ is a degenerate distribution for deterministic transition dynamics and the future trajectory $\boldsymbol{\tau} \sim p(s_{t+1}, s_{t+2}, \ldots | s_t)$ is strictly equal in distribution to $\boldsymbol{a} \sim \pi(a_t, a_{t+1}, \ldots | s_t)$, hence simplified as the latter in the following sections.

To learn the policy more efficiently, we let $s_t$ and $\boldsymbol{v}_t$ share a few mutual embedding layers, and provided option to pre-train the first $n_g$ layers with supervised learning. Layers inherited from pre-training are not updated during the training of RL. See Figure 2 for an overview of the architecture.

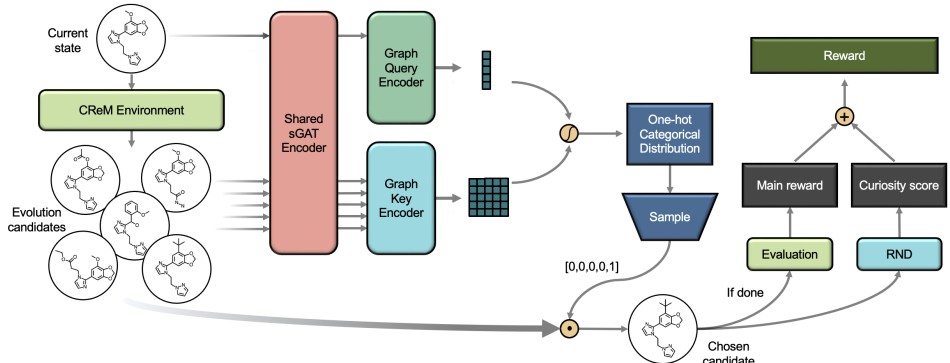

Figure 2: An overview of the Distilled Graph Attention Policy Network during a single step of the generating process.

### 2.3.1 ACTION SELECTION

At each time step $t$, we sample the next state $s_{t+1}$ from a categorical distribution constructed by applying a retrieval-system-inspired attention mechanism (Vaswani et al., 2017):

$$s_{t+1} \sim OHC \left\{ softmax \left( \bigcup_{g \in \boldsymbol{g}_{t+1}} \{L_{final}(E_Q(g_t) \parallel E_K(g))\} \right) \right\} \cdot \boldsymbol{v}_{t+1} \quad (5)$$

where $OHC\{p_1, \ldots, p_{n_v}\}$ is a one-hot categorical distribution with $n_v$ categories; $g_t$, $\boldsymbol{g}_{t+1}$ are the embeddings for $s_t$ and $\boldsymbol{v}_{t+1}$ acquired by the shared encoder; $E_Q$, $E_K$ are two sGAT+MLP graph encoders with output feature dimension $d_k$; $L_{final} : \mathbb{R}^{b \times 2d_k} \to \mathbb{R}^b$ is the final feed-forward layer. Essentially, each candidate state is predicted a probability based on its 'attention' to the query state. The next state is then sampled categorically according to these probabilities.

There could be a number of ways to determine stopping time $T$. For instance, an intuitive approach would be to append $s_t$ to $\boldsymbol{v}_{t+1}$ and terminate the process if $s_t$ is selected as $s_{t+1}$. In our experiments, we simply pick $T$ to be constant, i.e. we perform a fixed number of modifications for an input. This design encourages the process to not take meaningless long routes or get stuck in a cycle, and enables episodic docking evaluations in parallelization (further described in Section 2.5). Note that constant trajectory length is feasible because the maximum limit of time steps can be set significantly lower for fragment-based action space compared to node-by-node and edge-by-edge action spaces.

### 2.3.2 ACTOR-CRITIC ALGORITHM

For the purpose of obeying causal logic and reducing variance, the advantage on discounted reward-to-go are predominantly used instead of raw rewards in policy iterations. The Q-function and advantage function are expressed as

$$Q^\pi(s_t, a_t) = \mathbb{E}_\pi \left[ \sum_{t'=t}^{T} \gamma^{t'-t} \cdot r(s_{t'}, a_{t'}) \middle| s_t, a_t \right] \quad (6)$$

$$A^\pi(s_t, a_t) = Q^\pi(s_t, a_t) - \mathbb{E}_\pi \left[ Q^\pi(s_t, a_t) | s_t \right] \quad (7)$$

where $\gamma$ is the rate of time discount. The Advantage Actor-Critic (A2C) algorithm approximates $\mathbb{E}_\pi \left[ Q^\pi(s_t, a_t) | s_t \right]$ with a value network $V_\zeta(s_t)$ and $Q^\pi(s_t, a_t)$ with $r(s_t, a_t) + \gamma V_\zeta(s_{t+1})$. For a more detailed description of actor-critic algorithm in RL, see Grondman et al. (2012).

### 2.3.3 PROXIMAL POLICY OPTIMIZATION

We use Proximal Policy Optimization (PPO) (Schulman et al., 2017), a state-of-the-art policy gradient technique, to train our network. PPO holds a leash on policy updates whose necessity is elaborated in trust region policy optimization (TRPO) (Schulman et al., 2015), yet much simplified.

It also enables multiple epochs of minibatch updates within one iteration. The objective function is given as follow:

$$J^*(\theta) = \max_\theta \ \mathbb{E}_{\mathcal{D}, \pi_\theta^{old}} \left[ \sum_{t=1}^T \min \left\{ r_t(\theta) A^{\pi_\theta^{old}}(s_t, a_t), clip_\epsilon(r_t(\theta)) A^{\pi_\theta^{old}}(s_t, a_t) \right\} \right] \qquad (8)$$

where $r_t(\theta) = \pi_\theta^{new}(a_t|s_t)/\pi_\theta^{old}(a_t|s_t)$, $clip_\epsilon(x) = \min\{\max\{1 - \epsilon, x\}, 1 + \epsilon\}$ and $s_0 \sim \mathcal{D}$. During policy iterations, $\pi^{new}$ is updated each epoch and $\pi^{old}$ is cloned from $\pi^{new}$ each iteration.

## 2.4 Exploration with Random Network Distillation

We seek to employ a simple and efficient exploration technique that can be naturally incorporated into our architecture to enhance the curiosity of our policy. We perform Random Network Distillation (RND) (Burda et al., 2018) on graphs or pre-trained feature graphs to fulfill this need. Two random functions $\hat{f}_\psi, f^*$ that map input graphs to feature vectors in $\mathbb{R}^{d_r}$ are initialized with neural networks, and $\hat{f}_\psi$ is trained to match the output of $f^*$:

$$\psi^* = \arg\min_\psi \ \mathbb{E}_{s' \sim \hat{p}_{next}} \|\hat{f}_\psi(s') - f^*(s')\| \qquad (9)$$

where $\hat{p}_{next}$ is the empirical distribution of all the previously selected next states, i.e. the states that have been explored. We record the running errors in a buffer and construct the surrogate innovation reward as:

$$r_i(s') = clip_\eta \left( \left( \|\hat{f}_\psi(s') - f^*(s')\| - m_b \right) \Big/ \sqrt{v_b} \right) \qquad (10)$$

where $m_b$ and $v_b$ are the first and second central moment inferred from the running buffer, $clip_\eta(x) = \min\{\max\{-\eta, x\}, \eta\}$.

## 2.5 Parallelization and Synchronized Evaluation

Interacting with the environment and obtaining rewards through external software programs are the two major performance bottlenecks in ours as well as RL in general. An advantage of our environment settings, as stated in Section 2.3.1, is that a constant trajectory length is feasible. Moreover, the costs for environmental interactions are about the same for different input states. To take advantage of this, we parallelize environments on CPU subprocesses and execute batched operations on one GPU process, which enables synchronized and sparse docking evaluations that reduces the number of calls to the docking program. For future experiments where such conditions might be unrealistic, we also provided options for asynchronous Parallel-GPU and Parallel-CPU samplers (described in Stooke & Abbeel (2019)) in addition to the Parallel-GPU sampler used in our experiments.

# 3 Experiments

## 3.1 Setup

**Objectives**  We evaluated our model against five state-of-the-art models (detailed in Section 4) with the objective of discovering novel inhibitors targeting SARS-CoV-2 NSP15. **Molecular docking** scores are computed by docking programs that use the three-dimensional structure of the protein to predict the most stable bound conformations of the molecules of interest, targeting a pre-defined functional site. For more details on molecular docking and our GPU implementation of an automated docking tool used in the experiments, see Appendix B.2. In addition, we evaluated our model in the context of optimizing **QED** and **penalized LogP** values, two tasks commonly presented in machine learning literature for molecular design. The results for this can be found in Appendix D.

**Dataset**  For the models/settings that do require a dataset, we used a set of SMILES IDs taken from more than six million compounds from the MCULE molecular library — a publicly available dataset of purchasable molecules (Kiss et al., 2012), and their docking scores for the NSP15 target.

## 3.2 RESULTS

### 3.2.1 SINGLE-OBJECTIVE OPTIMIZATION

The raw docking score is a negative value that represents higher estimated binding affinity when the score is lower. We use the negative docking score as the main reward $r_m$ and assign it to the final state $s_T$ as the single objective. For DGAPN, we also assign innovation reward to each intermediate state, and the total raw reward for a trajectory $\tau$ is given by

$$r(\tau) = r_m(s_T) + \iota \cdot \sum_{t=1}^{T} r_i(s_t) \tag{11}$$

where $\iota$ is the relative important of innovation rewards, for which we chose 0.1 and incorporated them with a 100 episode delay and 1,000 episode cutoff. Detailed hyperparameter settings for DGAPN can be found in Appendix C. We sampled 1,000 molecules from each method and showed the evaluation results in Table 1. We note that we have a separate approach to evaluate our model that is able to achieve a $-7.73$ mean and $-10.38$ best docking score (see the **Ablation Study** paragraph below), but here we only evaluated the latest molecules found in training in order to maintain consistency with the manner in which GCPN and MolDQN are evaluated.

Table 1: Primary objective and other summary metrics in single-objective optimization

|  | Docking Score | | | | Validity | | Uniq. | Div. | QED | SA |
|---|---|---|---|---|---|---|---|---|---|---|
|  | 1st | 2nd | 3rd | mean | ordinary | adjusted |  |  |  |  |
| REINVENT | -8.38 | -7.33 | -7.28 | -4.66 | 95.1% | 94.6% | 95.1% | 0.88 | 0.64 | 7.51 |
| JTVAE | -7.48 | -7.32 | -6.96 | -4.55 | **100%** | **100%** | 34.2% | 0.87 | 0.67 | 2.65 |
| GCPN | -6.34 | -6.28 | -6.24 | -3.18 | **100%** | 78.6% | 100% | 0.91 | 0.47 | 5.48 |
| MolDQN | -8.01 | -7.92 | -7.86 | -5.38 | **100%** | **100%** | 100% | 0.88 | 0.37 | 5.04 |
| MARS | -8.11 | -7.99 | -7.86 | -5.11 | **100%** | 99.0% | 100% | 0.89 | 0.34 | 3.38 |
| DGAPN | **-10.07** | **-9.83** | **-9.19** | **-6.77** | **100%** | 99.8% | 100% | 0.81 | 0.31 | 2.91 |

In the result table, ordinary validity is checked by examining atoms' valency and consistency of bonds in aromatic rings. In addition, we propose adjusted validity which further deems molecules that fail on conformer generation (Riniker & Landrum, 2015) invalid on top of the ordinary validity criteria. This is required for docking evaluation, and molecules that fail this check are assigned a docking score of 0. We also provide additional summary metrics to help gain perspective of the generated molecules: Uniq. and Div. are the uniqueness and diversity (Polykovskiy et al., 2020); QED (Bickerton et al., 2012) is an indicator of drug-likeness, SA (Ertl & Schuffenhauer, 2009) is the synthetic accessibility. QED is better when the score is higher and SA is better when lower. Definitions of QED and SA can be found in Appendix E. On this task, DGAPN significantly outperformed state-of-the-art models in terms of top scores and average score, obtaining a high statistical significance over the second best model (MolDQN) with a p-value of $8.55 \times 10^{-209}$ under Welch's t-test (Welch, 1947). As anticipated, the molecules generated by fragment-based algorithms (JTVAE, MARS and DGAPN) have significantly better SAs. Yet we note that additional summary metrics are not of particular interest in single-objective optimization, and obtaining good summary metrics does not always indicate useful results. For example, during model tuning, we found out that worse convergence often tend to result in better diversity score. There also seems to be a trade-off between docking score and QED which we further examined in Section 3.2.3.

**Ablation study** We performed some ablation studies to examine the efficacy of each component of our model. Firstly, we segregated spatial graph attention from the RL framework and examined its effect solely in a supervised learning setting with the NSP15 dataset. The loss curves are shown in Figure 3, in which spatial convolution exhibited a strong impact on molecular graph representation learning. Secondly, we ran single-objective optimization with (DGAPN) and without (GAPN) innovation rewards, and thirdly, compared the results from DGAPN in evaluation against greedy algorithm with only the CReM environment. These results are shown in Table 2. Note that it is not exactly fair to compare greedy algorithm to other approaches since it has access to more information (docking reward for each intermediate candidate) when making decisions, yet our model still managed to outperform it in evaluation mode (see Appendix C for more information). From results

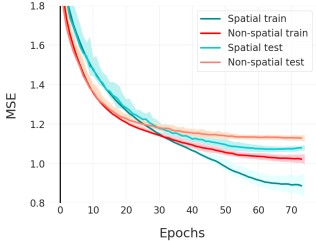

| | Docking Score | | | | QED | SA |
|---|---|---|---|---|---|---|
| | 1st | 2nd | 3rd | mean | | |
| GAPN | -9.19 | -8.96 | -8.90 | -6.71 | 0.28 | 2.78 |
| DGAPN | **-10.07** | **-9.83** | **-9.19** | **-6.77** | 0.31 | 2.91 |
| CReM Greedy | -9.65 | -9.36 | -9.36 | **-7.73** | 0.39 | 2.83 |
| DGAPN -eval | **-10.38** | **-9.84** | **-9.81** | **-7.73** | 0.34 | 3.03 |

Figure 3: Loss curves over 40 runs each with sample size 100,000

Table 2: Docking scores and other metrics under different training and evaluation settings

generated by the greedy approach, we can see that the environment and the stochasticity design of action space alone are powerful for the efficacy and exploration of our policies. While the innovation bonus helped discover molecules with better docking scores, it also worsened SA. We further investigated this docking score vs. SA trade-off in Section 3.2.3. To see samples of molecules generated by DGAPN in evaluation, visit our repository[†].

### 3.2.2 Constrained Optimization

The goal of constrained optimization is to find molecules that have large improvement over a given molecule from the dataset while maintaining a certain level of similarity:

$$r_{m'}(s_T) = r_m(s_T) - \lambda \cdot \max\{0, \delta - SIM\{s_0, s_T\}\} \tag{12}$$

where $\lambda$ is a scaling coefficient, for which we chose 100; $SIM\{\cdot, \cdot\}$ is the Tanimoto similarity between Morgan fingerprints. We used a subset of 100 molecules from our dataset as the starting molecules, chose the two most recent and best performing benchmark models in single-objective optimization to compete against, and evaluated 100 molecules generated from theirs and ours. The results are shown in Table 3.

Table 3: Objective improvements and molecule similarities under different constraining coefficients

| | MolDQN | | MARS | | DGAPN | |
|---|---|---|---|---|---|---|
| $\delta$ | Improvement | Similarity | Improvement | Similarity | Improvement | Similarity |
| 0 | $0.62 \pm 0.86$ | $0.22 \pm 0.06$ | $0.10 \pm 1.50$ | $0.15 \pm 0.06$ | $\mathbf{1.69} \pm 1.35$ | $\mathbf{0.26} \pm 0.18$ |
| 0.2 | $0.58 \pm 0.91$ | $0.26 \pm 0.07$ | $0.02 \pm 1.30$ | $0.16 \pm 0.07$ | $\mathbf{1.45} \pm 1.05$ | $\mathbf{0.32} \pm 0.21$ |
| 0.4 | $0.25 \pm 0.95$ | $\mathbf{0.46} \pm 0.09$ | $-0.06 \pm 1.20$ | $0.16 \pm 0.06$ | $\mathbf{0.41} \pm 1.06$ | $0.42 \pm 0.18$ |
| 0.6 | $0.01 \pm 0.75$ | $\mathbf{0.67} \pm 0.16$ | $0.09 \pm 1.23$ | $0.17 \pm 0.06$ | $\mathbf{0.33} \pm 0.68$ | $0.64 \pm 0.21$ |

From the results, it seems that MARS is not capable of performing optimizations with similarity constraint. Compared to MolDQN, DGAPN gave better improvements across all levels of $\delta$, although MolDQN was able to produce molecules with more stable similarity scores.

### 3.2.3 Multi-objective Optimization

We investigate the balancing between main objective and realism by performing multi-objective optimization, and thus provide another approach to generate useful molecules in practice. We weight $r_m$ with two additional metrics — QED and SA, yielding the new main reward as

$$r_{m'}(s_T) = \omega \cdot r_m(s_T) + (1 - \omega) \cdot \mu \cdot [QED(s_T) + SA^*(s_T)] \tag{13}$$

where $SA^*(s_T) = (10 - SA(s_T))/9$ is an adjustment of SA such that it ranges from 0 to 1 and is preferred to be larger; $\mu$ is a scaling coefficient, for which we chose 8. The results obtained by DGAPN under different settings of $\omega$ are shown in Figure 4. With $\omega = 0.6$, DGAPN is able to generate molecules having better average QED (0.72) and SA (2.20) than that of the best model (JTVAE) in terms of these two metrics in Table 1, while still maintaining a mean docking score ($-5.69$) better than all benchmark models in single-objective optimization.

---

[†]https://github.com/yulun-rayn/DGAPN

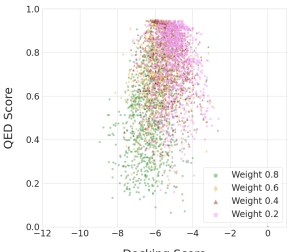 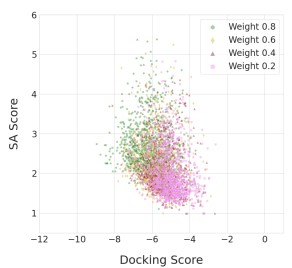 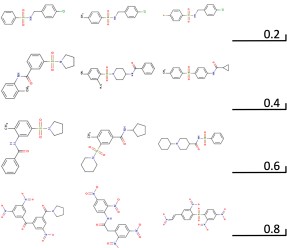

Figure 4: Summary of the molecules obtained by DGAPN. Left and middle plots show QED and SA vs. docking scores for the latest 1,000 molecules generated under each weight $\omega$. Right plot shows the top 3 molecules (out of 10,000) with the highest multi-objective rewards under each $\omega$.

A trade-off between docking reward and QED/SA was identified. We acknowledge that optimizing docking alone does not guarantee finding practically useful molecules, but our goal is to generate promising chemicals with room for rational hit optimization. We also note that commonly used alternative main objectives such as pLogP and QED are themselves unreliable or undiscerning as discussed in Appendix D. Hence, for methodological study purposes, we believe that molecular docking provides a more useful and realistic test bed for algorithm development.

## 4 RELATED WORK

The **REINVENT** (Olivecrona et al., 2017) architecture consists of two recurrent neural network (RNN) architectures, generating molecules as tokenized SMILE strings. The "Prior network" is trained with maximum likelihood estimation on a set of canonical SMILE strings, while the "Agent network" is trained with policy gradient and rewarded using a combination of task scores and Prior network estimations. The Junction Tree Variational Autoencoder (**JTVAE**, Jin et al. (2018)) trains two encoder/decoder networks in building a fixed-dimension latent space representation of molecules, where one network captures junction tree structure of molecules and the other is responsible for fine grain connectivity. Novel molecules with desired properties are then generated using Bayesian optimization on the latent space. Graph Convolutional Policy Network (**GCPN**, You et al. (2018a)) is a policy gradient RL architecture for de novo molecular generation. The network defines domain-specific modifications on molecular graphs so that chemical validity is maintained at each episode. Additionally, the model optimizes for realism with adversarial training and expert pre-training using trajectories generated from known molecules in the ZINC library. Molecule Deep Q-Networks (**MolDQN**, Zhou et al. (2019)) is a Q-learning model using Morgan fingerprint as representations of molecules. To achieve molecular validity, chemical modifications are directly defined for each episode. To enhance exploration of chemical space, MolDQN learns $H$ independent Q-functions, each of which is trained on separate sub-samples of the training data. Markov Molecular Sampling (**MARS**, Xie et al. (2021)) generates molecules by employing an iterative method of editing fragments within a molecular graph, producing high-quality candidates through Markov chain Monte Carlo sampling (MCMC). MARS then uses the MCMC samples in training a GNN to represent and select candidate edits, further improving sampling efficiency.

## 5 CONCLUSIONS

In this work, we introduced a spatial graph attention mechanism and a curiosity-driven policy network to discover novel molecules optimized for targeted objectives. We identified candidate antiviral compounds designed to inhibit the SARS-CoV-2 protein NSP15, leveraging extensive molecular docking simulations. Our framework advances the state-of-the-art algorithms in the optimization of molecules with antiviral potential as measured by molecular docking scores, while maintaining reasonable synthetic accessibility. We note that a valuable extension of our work would be to focus on lead-optimization — the refinement of molecules already known to bind the protein of interest through position-constrained modification. Such knowledge-based and iterative refinements may help to work around limitations of the accuracy of molecular docking predictions.

ACKNOWLEDGMENTS

This work was funded via the DOE Office of Science through the National Virtual Biotechnology Laboratory (NVBL), a consortium of DOE national laboratories focused on the response to COVID-19, with funding provided by the Coronavirus CARES Act. This research used resources of the Oak Ridge Leadership Computing Facility (OLCF) at the Oak Ridge National Laboratory, which is supported by the Office of Science of the U.S. Department of Energy under Contract No. DE-AC05-00OR22725. This manuscript has been coauthored by UT-Battelle, LLC under contract no. DE-AC05-00OR22725 with the U.S. Department of Energy. The United States Government retains and the publisher, by accepting the article for publication, acknowledges that the United States Government retains a nonexclusive, paid-up, irrevocable, world-wide license to publish or reproduce the published form of this manuscript, or allow others to do so, for United States Government purposes. The Department of Energy will provide public access to these results of federally sponsored research in accordance with the DOE Public Access Plan (http://energy.gov/downloads/doe-public-access-plan, last accessed September 16, 2020).

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

## APPENDIX

## A  DETAILED FORMULATION OF THE PROBLEM

Our goal is to establish a set of decision rules to generate graph-structured data that maximizes compound objectives under certain constraints. Similar to prior formulations, the generating process is defined as a time homogeneous Markov Decision Process (MDP). We give a formal definition of this process in Appendix A.1. Under this setting, the action policies and state transition dynamics at step $t$ can be factorized according to the Markov property:

$$P(a_t|s_0, a_0, s_1, a_1, \ldots, s_t) = P(a_t|s_t) := \pi(a_t|s_t) \tag{14}$$

$$P(s_{t+1}|s_0, a_0, s_1, a_1, \ldots, s_t, a_t) = P(s_{t+1}|s_t, a_t) := \rho(s_{t+1}|s_t, a_t) \tag{15}$$

where $\{s_t, a_t\}_t$ are state-action sequences. A reward function $r(s, a)$ is used to assess an action $a$ taken at a given state $s$. The process terminates at an optional stopping time $T$ and $s_T$ is then proposed as the final product of the current generating cycle. We aim to estimate the optimal policy $\pi$ in terms of various objectives to be constructed later in the experiment section.

### A.1 MEASURE THEORY CONSTRUCTION OF MARKOV DECISION PROCESS

Let $(S, \mathcal{S})$ and $(A, \mathcal{A})$ be two measurable spaces called the state space and action space; functions $\Pi : S \times \mathcal{A} \to \mathbb{R}$ and $T : S \times A \times \mathcal{S} \to \mathbb{R}$ are said to be a policy and a transition probability respectively if

1. For each $s \in S$, $E \to \Pi(s, E)$ is a probability measure on $(A, \mathcal{A})$; for each $(s, a) \in S \times A$, $F \to T(s, a, F)$ is a probability measure on $(S, \mathcal{S})$.

2. For each $E \in \mathcal{A}$, $s \to \Pi(s, E)$ is a measurable function from $(S, \mathcal{S}) \to (\mathbb{R}, \mathcal{B})$; for each $F \in \mathcal{S}$, $(s, a) \to T(s, a, F)$ is a measurable function from $(S \times A, \mathcal{S} \otimes \mathcal{A}) \to (\mathbb{R}, \mathcal{B})$.

We say a sequence of random variable duples $(S_t, A_t)$ defined on the two measurable spaces is a Markov decision chain if

$$P(A_t \in E \mid \sigma(S_0, A_0, S_1, A_1, \ldots, S_t)) = \Pi(S_t, E) \tag{16}$$

$$P(S_{t+1} \in F \mid \sigma(S_0, A_0, S_1, A_1, \ldots, S_t, A_t)) = T(S_t, A_t, F) \tag{17}$$

A function $r : S \times \mathcal{A} \to \mathbb{R}$ is said to be the reward function w.r.t. the Markov decision chain if $r(s_t, E_t) = \mathbb{E}_{\Pi, T} [R(s_{t+1}) \mid S_t = s_t, A_t \in E_t]$ where $R : S \to \mathbb{R}$ is its underlying reward function.

With an abuse of notation, we define $\pi(a|s) := \Pi(s, \{a\})$, $\rho(s'|s, a) := T(s, a, \{s'\})$ and let $r(s, a)$ denote $r(s, \{a\})$.

## B   LEARNING ENVIRONMENT AND REWARD EVALUATION

### B.1   ENVIRONMENT - CREM

Chemically Reasonable Mutations (CReM) is an open-source fragment-based framework for chemical structure modification. The use of libraries of chemical fragments allows for a direct control of the chemical validity of molecular substructures and to consider the chemical context of coupled fragments (e.g., resonance effects).

Compared to atom-based approaches, CReM explores less of chemical space but guarantees chemical validity for each modification, because only fragments that are in the same chemical context are interchangeable. Compared to reaction-based frameworks, CReM enables a larger exploration of chemical space but may explore chemical modifications that are less synthetically feasible. Fragments are generated from the ChEMBL database (Gaulton et al., 2012) and for each fragment, the chemical context is encoded for several context radius sizes in a SMILES string and stored along with the fragment in a separate database. For each query molecule, mutations are enumerated by matching the context of its fragments with those that are found in the CReM fragment-context database (Polishchuk, 2020).

In this work, we use grow function on a single carbon to generate initial choices if a warm-start dataset is not provided, and mutate function to enumerate possible modifications with the default context radius size of 3 to find replacements.

### B.2   EVALUATION - AUTODOCK-GPU

Docking programs use the three-dimensional structure of the protein (i.e., the receptor) to predict the most stable bound conformations of the small molecules (i.e., its putative ligands) of interest, often targeting a pre-defined functional site, such as the catalytic site. An optimization algorithm within a scoring function is employed towards finding the ligand conformations that likely correspond to binding free energy minima. The scoring function is conformation-dependent and typically comprises physics-based empirical or semi-empirical potentials that describe pair-wise atomic terms, such as dispersion, hydrogen bonding, electrostatics, and desolvation (Huang et al., 2010; Huey et al., 2007). AutoDock is a computational simulated docking program that uses a Lamarckian genetic algorithm to predict native-like conformations of protein-ligand complexes and a semi-empirical scoring function to estimate the corresponding binding affinities. Lower values of docking scores indicate stronger predicted interactions. The opposite value of the lowest estimated binding affinity energy obtained for each molecule forms the reward.

AutoDock-GPU (Santos-Martins et al., 2021) is an extension of AutoDock to leverage the highly-parallel architecture of GPUs. Within AutoDock-GPU, ADADELTA (Zeiler, 2012), a gradient-based method, is used for local refinement. The structural information of the receptor (here, the NSP15 protein) used by AutoDock-GPU is processed prior to running the framework. In this preparatory step, AutoDockTools (Morris et al., 2009b) was used to define the search space for docking on NSP15 (PDB ID 6W01; Figure 5) and to generate the PDBQT file of the receptor, which contains atomic coordinates, partial charges, and AutoDock atom types. AutoGrid4 (Morris et al., 2009a) was used to pre-calculate grid maps of interaction energy at the binding site for the different atom types defined in CReM.

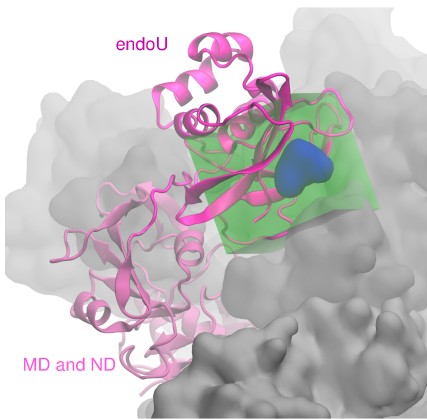

Figure 5: The search space in NSP15 defined for molecular docking (green box). An NSP15 protomer, which was used as the receptor in the calculations, is shown (cartoon backbone representation, in pink/magenta). The nucleotide density located at the catalytic site is depicted (blue surface). Other protomers forming the homo-hexamer are shown as grey surfaces. PDB IDs 6WLC and 6WXC were used in this illustration (Kim et al., 2021). Abbreviations: EndoU, Poly-U specific endonuclease domain; MD, Middle domain; ND, N-terminal domain.

In evaluation, after applying an initial filter within RDKit to check whether a given SMILES is chemically valid (i.e., hybridization, ring membership etc.), a 3D conformer of the molecule is generated using AllChem.EmbedMolecule. SMILES that do not correspond to valid compounds are discarded. Next, the molecular geometry is energy minimized within RDKit using the generalized force filed MMFF94. The resulting conformer is used as input for molecular docking via AutoDock-GPU. We also excluded any molecules from the final result set that were both fully rigid and larger than the search box in the receptor. This only occurred in two molecules from the JTVAE evaluation.

## C   HYPERPARAMETER SETTINGS FOR SINGLE-OBJECTIVE OPTIMIZATION

Based on a parameter sweep, we set number of GNN layers to be 3, MLP layers to be 3, with 3 of the GNN layers and 0 of the MLP layers shared between query and key. Number of layers in RND is set to 1; all numbers of hidden neurons 256; learning rate for actor $2^{-3}$, for critic $1^{-4}$, for RND $2^{-3}$; update time steps (i.e. batch size) 300. Number of epochs per iteration and clipping parameter $\epsilon$ for PPO are 30 and 0.1. Output dimensions and clipping parameter $\eta$ for RND are 8 and 5. In evaluation mode, we use $\arg\max$ policy instead of sampling policy, expand the number of candidates per step from 15-20 to 128 and expand the maximum time steps per episode from 12 to 20 compared to training. For more details regarding hyperparameter settings, see our codebase at https://github.com/yulun-rayn/DGAPN.

## D   MORE RESULTS ON QED AND PENALIZED LOGP

Although QED and penalized LogP are the most popular objectives to benchmark ML algorithms for molecule generation, these benchmarks are questionable for both scientific study and practical use as Xie et al. (2021) pointed out. Most methods can obtain QED scores close or equal to the

highest possible of 0.948, making the metric hard to distinguish different methods. As for pLogP, if we simply construct a large molecule with no ring, such as the molecule from SMILES 'CC-CCC...CCCCC' (139 carbons), it will give us a pLogP score of 50.31 which beats all state-of-the-art models in Table 4. Needless to say, we will achieve a even higher pLogP by continuously adding carbons, which was exactly how REINVENT performed in our experiment. We note that we were able to raise our results to around 18 solely by doubling the maximum time steps per episode reported in Appendix C, yet not so interested in pushing the performance on this somewhat meaningless metric by continuously increasing one hyperparameter.

Table 4: Top QED and pLogP scores

|  | QED | | | pLogP | | |
| --- | --- | --- | --- | --- | --- | --- |
|  | 1st | 2nd | 3rd | 1st | 2nd | 3rd |
| REINVENT | 0.945 | 0.944 | 0.942 | **49.04** | **48.43** | **48.43** |
| JTVAE | 0.925 | 0.911 | 0.910 | 5.30 | 4.93 | 4.49 |
| GCPN | **0.948** | 0.947 | 0.946 | 7.98 | 7.85 | 7.80 |
| MolDQN | **0.948** | 0.944 | 0.943 | 11.84 | 11.84 | 11.82 |
| MARS | **0.948** | **0.948** | **0.948** | 44.99 | 44.32 | 43.81 |
| DGAPN | **0.948** | **0.948** | **0.948** | 12.35 | 12.30 | 12.22 |

The results from REINVENT were produced in our own experiments, while others were directly pulled out from the original results reported in the literature.

# E  DEFINITIONS OF QED AND SA

## E.1  QUANTITATIVE ESTIMATE OF DRUGLIKENESS

(QED) is defined as

$$QED = \exp\left(\frac{1}{n}\sum_{i=1}^{n}\ln d_i\right),$$

where $d_i$ are eight widely used molecular properties. Specifically, they are molecular weight (MW), octanol-water partition coefficient (ALOGP), number of hydrogen bond donors (HBD), number of hydrogen bond acceptors (HBA), molecular polar surface area (PSA), number of rotatable bonds (ROTB), the number of aromatic rings (AROM), and number of structural alerts. For each $d_i$,

$$d_i(x) = a_i + \frac{b_i}{1+\exp\left(-\frac{x-c_i+\frac{d_i}{2}}{e_i}\right)} \cdot \left[1 - \frac{1}{1+\exp\left(-\frac{x-c_i+\frac{d_i}{2}}{f_i}\right)}\right],$$

each $a_i, \ldots, f_i$ are given by a supplementary table in Bickerton et al. (2012).

## E.2  SYNTHETIC ACCESSIBILITY

(SA) is defined as

$$SA = \texttt{fragmentScore} - \texttt{complexityPenalty}$$

The fragment score is calculated as a sum of contributions from fragments of 934,046 PubChem already-synthesized chemicals. The complexity penalty is computed from a combination of ringComplexityScore, stereoComplexityScore, macroCyclePenalty, and the sizePenalty:

$$\texttt{ringComplexityScore} = \log(\texttt{nRingBridgeAtoms} + 1) + \log(\texttt{nSprioAtoms} + 1)$$
$$\texttt{stereoComplexityScore} = \log(\texttt{nStereoCenters} + 1)$$
$$\texttt{macroCyclePenalty} = \log(\texttt{nMacroCycles} + 1)$$
$$\texttt{sizePenalty} = \texttt{nAtoms}^{1.005} - \texttt{nAtoms}$$

