# OpenReview forum: "Spatial Graph Attention and Curiosity-driven Policy for Antiviral Drug Discovery"
_ICLR.cc/2022/Conference — ICLR 2022 Poster_

### Official Review · Reviewer_ce6h · 2021-10-29

**Correctness:** 4
**Technical Novelty And Significance:** 2
**Empirical Novelty And Significance:** 2
**Recommendation:** 6
**Confidence:** 3

**Main Review:**

Strengths：
1.	The model in the paper is designed to resolve the up-to-date challenge of discovering novel inhibitors to target the SARS-CoV-2 non-structural protein endoribonuclease, and the experimental results show that molecules generated by the proposed model enjoy better synthetic accessibility according to SA value, which is a metric generally used to measure the synthetic difficulty of drugs. Meanwhile, the molecules show higher docking scores compared to other existing algorithms.
2.	The reinforcement learning part of the model is based on a fragment-based chemical environment for chemical synthesis and utilizes many novel approaches like Proximal Policy Optimization and Random Network Distillation as an effective attempt. This may give inspirations of the model architecture to future researches in this field.

Weaknesses:
1.	Spatial graph attention and spatial convolution indeed improve the model’s ability to extract structural information from input graphs, but these modules are not innovative enough. Actually, spatial graph attention is just a multi-head attention mechanism applied to graphs and spatial convolution is a Euclidean distance version of GNN’s convolution operation. Some new modules are expected to be employed by the model to make it more powerful.
2.	As is seen from the experimental results, the docking scores and SA value are better than other existing algorithms indeed. But the model’s performance on other metrics is unsatisfactory and particularly, Diversity is even the worst among algorithms in Table 1. Although this has been explained in the paper as a general drawback of fragment-based algorithms, more optimizations may need to be realized.
3.	Some statements in the paper may make readers confused, which are listed below. Equation 3 and 4 don’t share a consistent representation of the element in the adjacency matrix. Table 1 lacks some necessary annotations like what is the definition of Diversity and what is the meaning of the values in bold. The right plot in Figure 4 also lacks some necessary explanations and this plot seems to be redundant now.


**Summary Of The Paper:**

With the goal of generating molecules that bind to functional sites of SARS-Cov-2 protein, the paper proposed a reinforcement learning model with a fragment-based framework for chemical structure modification. In the network part, spatial graph attention and spatial convolution are utilized to extract more structural information from the input graph into the representation of nodes or graphs. Based on the actor-critic algorithm, the reinforcement learning part is designed to find the state with the best docking score computed by docking programs and some novel approaches like PPO and RND are used to train the model more effectively. In the experiments, the model shows great performance while reducing the complexity of chemical synthesis meanwhile.

**Summary Of The Review:**

The paper utilized many novel approaches in graph representation and reinforcement learning to resolve the timely challenge about SARS-CoV-2, but the experimental results of the proposed model fail to show sufficiently powerful performance compared to existing algorithms except for better synthetic accessibility and docking scores. More innovative designs may need to be introduced to the present model to get a persuasive improvement.

---

> ### Author Response · Authors · 2021-11-21
> **Reply to Reviewer ce6h**
>
> Thanks for the review and comments. We understand that the primary concern of yours is that we demonstrate ‘sufficiently powerful performance compared to existing algorithms’. Here, we attempt to point to information in the paper that supports our method as state-of-the-art, and make heavy use of the reviewer’s comments to improve the quality of our manuscript.
>
> 1. ‘As is seen from the experimental results, the docking scores and SA value are better than other existing algorithms indeed. But the model’s performance on other metrics is unsatisfactory’
>
> First, we wish to clarify some of the information presented in Table 1. Specifically, the docking score is the **only** single objective we are optimizing. Additionally, the CReM environment was chosen to ensure chemical validity and reduce SA, but besides these three metrics, there is no reason for our model or any model to do well in the other metrics such as QED in the first place if it is not directly in the objective/reward function or not being specifically taken care of by the construction of action space. We note that most papers on the same subject only report validity alongside the main objective, other metrics were included in our paper only because we wish to help readers gain some summary knowledge of the generated samples from each model. If QED is of particular concern in practice, we will put it in the objective function, and as shown in Appendix C and the left plot of Figure 4. in Section 4.2.3 (multi-objective optimization), our model easily achieves the best QED 0.948 when it is either the sole objective or weighted with other properties in the objective function. We have modified the caption of Table 1, and briefly summarized the point above at the end of paragraph 1 in Section 4.2.1 in the revised version. We also modified the statement regarding the purpose of multi-objective optimization (first sentence of Section 4.2.3) in the manuscript to further clarify this point.
>
> As of the single objective, namely docking score, DGPAN outperformed all state-of-the-art models by a significant margin, indicated in Table 1, with a p-value of 1.67e-37 under Welch’s t-test over the second best algorithm MolDQN, and exhibited even a higher margin from ‘DGAPN eval’ in Table 2. We are convinced that this is a significant improvement on the state-of-the-art. We have added the t-stat into the revised version.
>
>
> 2. ‘Diversity is even the worst among algorithms in Table 1. Although this has been explained in the paper as a general drawback of fragment-based algorithms, more optimizations may need to be realized.’ ‘Table 1 lacks some necessary annotations like what is the definition of Diversity and what is the meaning of the values in bold.’
>
> We agree that the definition of diversity needs to be included. We added that to the rebuttal version (please see just below Table 1).
>
> We note that better convergence will naturally lead to lower diversity a lot of the time. For example, we have rerun MolDQN after the submission and it converged to a better mean of -7.1 while its diversity is reduced to 0.87. However, this score is still actionable, in the sense that many high-affinity molecules are returned, providing opportunities for lead optimization -- this, after all, is the principle use-case of such generative models. In binary terms, one can classify our diversity score as the ‘worst’, but except for GCPN (which is the one structure that wasn’t able to learn and converge in this problem), all models’ diversity scores are within the range of 0.86-0.88, meaning the ‘best’ is only 0.02 above ours. These models are all well within an actionable range and it is not particularly our goal to generate a set of molecules that is more diverse than that of the other models. In the revised revision, we updated the new results for MolDQN and briefly discussed this point (Section 4.2.1 end of first paragraph).
>
>
> 3. ‘Some statements in the paper may make readers confused, which are listed below. Equation 3 and 4 don’t share a consistent representation of the element in the adjacency matrix.’‘The right plot in Figure 4 also lacks some necessary explanations and this plot seems to be redundant now.’
>
> Thank you for this feedback to help improve and clarify our explanations. We have fixed the inconsistency you pointed out in Eq. 3 and 4., and gave a more precise description to the right plot in Fig. 4. - Each row is the top three molecules (in terms of the multi-objective score) in each setting of weight \omega.
>
> Summary: We appreciate your positive comment on the novelty of the framework and its inspiration to future works in the field, and have clarified the concerns you raised in the revised version as described above. We truly believe that this study merits publication in the ICLR as the results have shown important advancement and demonstrated sufficiently powerful performance compared to existing algorithms.

---

> > ### Comment · Reviewer_ce6h · 2021-11-21
> > **Reply to the authors**
> >
> > Thank you for your reply and I have read your clarifications and explanations for the questions I raised in my review. Now I understand you want to stress that the docking score is the only single objective to be optimized and from this perspective, your model is sufficiently powerful compared with existing algorithms. So I change my recommendation score to 6.
> >
> > But I still hold the belief that the modules proposed in the paper like spatial grpah attention and spatial convolution are not innovative enough. More improvement may need to be shown in your future work to design a stronger nerual network part inside your model. Meanwhile, I'm delighted to see that the confusing statements in the previous version has been modified and I strongly suggest that in your later work, this kind of check could be done before submission.

---

### Official Review · Reviewer_ZQrT · 2021-11-02

**Correctness:** 3
**Technical Novelty And Significance:** 3
**Empirical Novelty And Significance:** Not applicable
**Recommendation:** 6
**Confidence:** 4

**Main Review:**

Pros:
- The authors extend graph attention networks to edge features and notice that spatial information is crucial in activity prediction. The introduced sGAT model is a novel contribution. It would be interesting to see an ablation study comparing this model to simpler graph neural networks, e.g. to the closely related EGAT architecture [1].
- Chemically reasonable mutations (CReM) are sampled by the RL algorithm to ensure that the generated structures are realistic.
- An innovation reward is added to encourage exploration. The reward is based on random network distillation.
- The ablation study shows the usefulness of spatial convolution for the supervised learning task.
- The results of the pretraining are shown in the ablation study even though they do not improve the generated compounds in terms of the main objective (an interesting negative result).

Cons:
- The authors state that "fragment-by-fragment sequential generation [...] has not been utilized in a deep graph RL framework", but I am not sure it is true. For example, Ståhl et al. [2] use deep reinforcement learning on fragments to optimize molecules.
- Recently, more and more publications focus on optimizing molecular docking scores using de novo drug design models, and I think this topic should be briefly summarized in the related work. For example, see [3].
- In Figure 1, the meaning of lines, symbols, and colors is unclear. Even after reading Equations 3-6, the drawing is difficult to process.
- The mathematical notation is difficult to follow in some parts of the methods section. For example, what is the purpose of the OHC operator? Is it there only to sample from the softmax distribution and map the decision onto the set of valid molecules?
- The QED score is very low for DGAPN. Can it be caused by generating big compounds that exceed a typical molecular weight of drug-like compounds?  The compounds are generated by adding whole chemical fragments, which can cause fast growth of the molecules. If that is the case, then these structures will be difficult to optimize (cannot be easily used as hits, which was suggested by the authors). This defeats the purpose of the generative model. The generated structures should be shown at least in the supplementary material.
- In this paper, only one antiviral target is considered. The results would be more convincing if there were more proteins in the experiments.
- The rewards are based on docking scores, so it would be interesting to see the docking poses of generated compounds. Especially, large compounds like those shown in Figure 4, which probably barely fit in the binding pocket.

Other comments:
- Why is the second column for DGAPN bolded in Figure 1?
- In Section 5, there is a citation that failed to compile: "MARS, xie2021mars"
- In Appendix D.3, the notation $p_w$ is used two times - should it not be just $p$ for the generated molecules?

[1] Chen, Jun, and Haopeng Chen. "Edge-Featured Graph Attention Network." arXiv preprint arXiv:2101.07671 (2021).

[2] Ståhl, Niclas, et al. "Deep reinforcement learning for multiparameter optimization in de novo drug design." Journal of chemical information and modeling 59.7 (2019): 3166-3176.

[3] Thomas, Morgan, et al. "Comparison of structure-and ligand-based scoring functions for deep generative models: a GPCR case study." Journal of Cheminformatics 13.1 (2021): 1-20.

**Summary Of The Paper:**

The authors propose a new fragment-based method for molecule generation. The model, called DGAPN, uses chemical fragments extracted from a public compound library with their chemical context (atom neighborhoods to which these fragments are attached). This way, all modifications made by the model should produce synthetically accessible compounds with high probability. Additionally, the authors introduce a new class of graph neural networks to predict chemical properties that employ an attention mechanism over atoms and chemical bonds. These models are used to guide the generative MDP trained with reinforcement learning. Inhibition of a SARS-CoV-2 antiviral target, NSP15, is used as an example task for the proposed model. The experimental section shows the results of single- and multi-objective compound generation, in which DGAPN obtains better docking scores of the generated molecules than other methods.

**Summary Of The Review:**

For now, I recommend the rejection of this paper (weak reject). The novelty of the proposed method is significant, but there are some unclear parts in the description of the method, and the results do not sufficiently prove the superiority of the proposed approach (see the comments above).

---

> ### Author Response · Authors · 2021-11-21
> **Reply to Reviewer ZQrT**
>
> 1. ‘It would be interesting to see an ablation study comparing this model to simpler graph neural networks, e.g. to the closely related EGAT architecture’
>
> We agree, and have indeed done this. The comparison in Figure 3 is using sGAT vs. only using the attribution graph of sGAT (which is edge featured), which demonstrates the power of spatial attention.
>
>
> 2. ‘The authors state that "fragment-by-fragment sequential generation [...] has not been utilized in a deep graph RL framework", but I am not sure it is true. For example, Ståhl et al. [2] use deep reinforcement learning on fragments to optimize molecules’
>
> Ståhl et al. seem to use binary string structures to represent molecules instead of graph structures. To the best of our knowledge, our statement referenced above is true.
>
>
> 3. ‘The mathematical notation is difficult to follow in some parts of the methods section. For example, what is the purpose of the OHC operator? Is it there only to sample from the softmax distribution and map the decision onto the set of valid molecules?’
>
> Yes. Formally, the purpose of OHC is to relate the action selection mechanism to the Q,K,V mechanism more closely, in which the vector inferred by query Q and key K is operated in dot product with the values V, i.e. the molecules. Hence why it is written in this form. In the revised version, we have provided more intuitive explanations for these equations (page 5 end of first paragraph).
>
>
> 4. ‘The QED score is very low for DGAPN. Can it be caused by generating big compounds that exceed a typical molecular weight of drug-like compounds? The compounds are generated by adding whole chemical fragments, which can cause fast growth of the molecules. If that is the case, then these structures will be difficult to optimize (cannot be easily used as hits, which was suggested by the authors). This defeats the purpose of the generative model. The generated structures should be shown at least in the supplementary material.’ & ‘the results do not sufficiently prove the superiority of the proposed approach’
>
> In single-objective setting, we did find that some of the structures the model generates have repeatedly added particular fragments that would help maximize the interaction with the receptor. This can sometimes cause a lower QED. However, we emphasize that the primary purpose of the single-objective optimization is to examine the power of our framework in terms of exploiting this sole objective, while the practical problem is addressed in the multi-objective setting, where we add weight to auxiliary parameters, such as QED (Section 4.2.3.). This protocol favors the generation of simpler molecules that leaves room for rational optimization. But in terms of the single objective, the docking score, there is a significant improvement in ours with a p-value on the scale of 10^-37 compared to the second best model (MolDQN) in Table 1 under Welch’s t-test, and an even higher margin for DGAPN eval in Table 2. These results indicate the superiority of our proposed approach on this task.
>
> We did submit a file in supplementary material of 500 sampled molecules from DGAPN in single-objective optimization, and a link to this file will be provided in the appendix in the final version.
>
>
> 5. ‘In this paper, only one antiviral target is considered. The results would be more convincing if there were more proteins in the experiments.’
>
> We had data from an alternative binding pocket in nsp15, which would be technically equivalent to using a different protein target. Similar improvements were obtained for this different target, however, those were not included for two reasons. First, we sought advice from our biologists, who pointed out that this alternative binding site is unlikely to be functionally relevant. More importantly, it might not actually contribute to a significant increase in the confidence of the method among readers/reviewers because they might want to see the model works under a different context. Hence why we tested and showed the results of DGAPN on the more standard common objectives like PlogP and QED in addition to the docking reward (Appendix C), instead of showing its performance on another functional site or protein.
>
>
> Summary: Overall, we thank the reviewer for comments that enabled us to better clarify that our results provide important gains for drug design methodology compared to the state-of-the-art, and we note that fragment-by-fragment sequential generation has indeed not been utilized in a deep graph RL framework to the best of our knowledge. Further, our ablation analyses indicate that the novel components of our design are indeed important for our state-of-the-art performance. Due to character limit, we cannot list all the changes in the revised version here, but we have made improvements to the manuscript according to your other concerns as well. We hope that you will agree that our revisions and responses enable publication in ICLR.

---

> > ### Comment · Reviewer_ZQrT · 2021-11-28
> > **Reply**
> >
> > Thank you for your response and explanations. Most of my comments were addressed, and I see work done to improve the manuscript. I looked through the generated molecules, and they indeed are on the heavy side, with many repeated substructures containing nitrogen chains. However, I do not think this is a significant drawback of the method as those are the results of the single-objective optimization. I am glad that the link to these results will be provided in the final version.
> >
> > After reading the authors' response and other reviews, I decided to change my score to 6.

---

### Official Review · Reviewer_wuts · 2021-11-05

**Correctness:** 3
**Technical Novelty And Significance:** 3
**Empirical Novelty And Significance:** 3
**Recommendation:** 8
**Confidence:** 4

**Main Review:**

Strengths:

Relevant application of designing small molecules with desired properties, eg inhibitors of SARS-CoV-2

Paper is overall well written

Proposed method compared with quite a large range of relevant baselines

Some informative ablation studies done

Code has been provided

Other comments/questions:

Although the proposed method is apparently the first to utilize a fragment by fragment graph construction in a deep reinforcement learning framework, I think the idea to extend the atom by atom graph construction in previous RL works with a fragment based action space (explored in more recent works Jin 2018, Jin 2020, Xie 2021) is interesting but not particularly novel.

Parts of figure 4, such as the molecular structures, are too small and unreadable

It would be useful to see the distribution of the properties in Table 1 (dock score, QED, SA, FCD). Also, it would be interesting to see the QED, SA, and FCD that correspond to the top 3 dock score molecules


**Summary Of The Paper:**

This paper proposes a method to generate molecular graphs with optimized properties. Molecular graphs are constructed by the iterative addition of molecular fragments in a deep reinforcement learning framework. The method is benchmarked against a set of baselines in a task to generate molecules that maximizes the docking score to a protein (NSP15) from the SARS-CoV-2 virus, and shows good performance.

**Summary Of The Review:**

Overall, I vote for acceptance. The well-structured paper proposes a useful molecule graph generation approach, accompanied by quite a comprehensive set of experiments.

---

> ### Author Response · Authors · 2021-11-21
> **Reply to Reviewer wuts**
>
> 1. 'This paper proposes a method to generate molecular graphs with optimized properties. Molecular graphs are constructed by the iterative addition of molecular fragments in a deep reinforcement learning framework. The method is benchmarked against a set of baselines in a task to generate molecules that maximizes the docking score to a protein (NSP15) from the SARS-CoV-2 virus, and shows good performance.'
>
> We thank the reviewer for the positive and accurate summary.
>
>
> 2. 'Relevant application of designing small molecules with desired properties, eg inhibitors of SARS-CoV-2; Paper is overall well written; Proposed method compared with quite a large range of relevant baselines; Some informative ablation studies done; Code has been provided'
>
> We thank the reviewer; we endeavored to present a comprehensive set of comparisons to other methods, and to use ablation studies to provide insight into which aspects of our algorithm are responsible for improved performance benchmarks.
>
>
> 3. 'Although the proposed method is apparently the first to utilize a fragment by fragment graph construction in a deep reinforcement learning framework, I think the idea to extend the atom by atom graph construction in previous RL works with a fragment based action space (explored in more recent works Jin 2018, Jin 2020, Xie 2021) is interesting but not particularly novel.'
>
> We agree that the method has been suggested elsewhere. The library of fragments and reactions that we use is, to the best of our knowledge, by far the largest such library in the public domain.
>
>
> 4. 'Parts of figure 4, such as the molecular structures, are too small and unreadable'
>
> We have made the molecular structures thicker and more readable in the revised manuscript.
>
>
> 5. 'It would be useful to see the distribution of the properties in Table 1 (dock score, QED, SA, FCD). Also, it would be interesting to see the QED, SA, and FCD that correspond to the top 3 dock score molecules'
>
> We agree that it would be interesting to see, yet we try not to put too much emphasis on the joint distribution of docking scores and other metrics at the stage of single-objective optimization. None of the other metrics are being optimized at this stage and aside from validity and SA, other metrics were not of particular concern when building the framework and thus are not of particular concern if they are not included in the objective function. We included them just to give readers some perspectives of the molecules each model generates without having to draw out all of them. However, we did put a lot more emphasis on this matter when it comes to multi-objective optimization, where two of these metrics are actually in the objective and of interest. The distributions of them can be found in the left and middle plot of Figure 4.
>
> Besides, we also considered that using these metrics is not straightforward: for example, 25% of all FDA approved drugs have QED scores less than 0.4, and 10% have scores less 0.2 – many have scores of 0 (Kosugi & Ohue. Int. J. Mol. Sci. 2021, 22, 10925).
>
>
> 6. 'Summary Of The Review: Overall, I vote for acceptance. The well-structured paper proposes a useful molecule graph generation approach, accompanied by quite a comprehensive set of experiments.'
>
> We thank the reviewer for the useful comments, which have improved the clarity and quality of the revised manuscript.

---

> > ### Comment · Reviewer_wuts · 2021-11-30
> > **Reply**
> >
> > I thank the authors for their comments and leave my final scores unchanged

---

### Public Comment · ~Mikaela_Cashman1 · 2022-05-11
**Final Revision**

We thank the reviewers again for their helpful and generous feedback and the ICLR conference and attendees for their comments. We also acknowledge Benson Chen from Massachusetts Institute of Technology for bringing to our attention tool version differences.

As such, we have updated our paper and workflow to reflect these comments, including using more recent versions of AutoDock-GPU and RDKit, as specified on our github. Furthermore, we have refined our method of molecular filtering prior to molecular docking (page 15) and that reduced the prevalence of nitrogen rich molecules pointed out by Reviewer ZQrT. Please note that, although absolute resulting values have changed, our discussion and conclusions were not affected.

-ICLR 2022 Conference Paper1126 Authors

---

### Decision · Program_Chairs · 2022-01-20

**Decision:**

Accept (Poster)

**Comment:**

After discussion, all reviewers are convinced about the novelty of the proposed method, and adjusted scores to recommend acceptance. They all appreciate the attempt to attack COVID-19 using machine learning.